# Physiological Ageing of the Lumbar Intervertebral Disc Based on Magnetic Resonance Imaging, a Systematic Literature Review

**DOI:** 10.3390/medicina61081430

**Published:** 2025-08-08

**Authors:** Max H. E. de Vries, Inge J. M. H. Caelers, Wouter L. W. van Hemert, Toon F. M. Boselie, Henk van Santbrink

**Affiliations:** 1Department of Orthopaedic Surgery, Zuyderland Medical Centre, 6162 BG Sittard-Geleen, The Netherlands; w.vanhemert@zuyderland.nl; 2Department of Neurosurgery, Zuyderland Medical Centre, 6162 BG Sittard-Geleen, The Netherlands; inge.caelers@mumc.nl (I.J.M.H.C.); t.boselie@mumc.nl (T.F.M.B.); h.van.santbrink@mumc.nl (H.v.S.); 3Department of Neurosurgery, Maastricht University Medical Centre, 6229 HX Maastricht, The Netherlands

**Keywords:** lumbar spine, (inter)vertebral disc, degeneration, magnetic resonance imaging, systematic review, physiology

## Abstract

*Background and Objectives:* All intervertebral discs (IVDs) degenerate with the progression of age. Currently we are unable to differentiate physiological lumbar intervertebral disc degeneration (LIDD) from pathophysiological using imaging. The first step in differentiating physiological from pathophysiological degeneration is to determine physiological LIDD. Biochemical and histological analysis are not viable tools to look at the IVD in patients or healthy subjects due to their invasive character. For this reason, a non-invasive MRI could be the solution to study the lumbar IVD and LIDD. Therefore, the purpose of this systematic literature review is to identify the physiological aging process of the lumbar IVD based on MRI studies in patients of all ages without a history of spine pathology or surgery. *Materials and Methods:* After searching four databases (PubMed, Embase, Web of Science, and Cochrane), titles and abstracts of the identified studies were screened using inclusion and exclusion criteria. Eligible articles were subjected to a full-text review. Quality assessment was performed using ROBINS-I for risk of bias and Oxford level of evidence. *Results*: In total, 38 articles were included in this review. Most studies were non-consecutive studies (*n* = 36). Two studies were exploratory cohort studies. Twenty-two studies were prospective studies and sixteen were retrospective. Level of evidence ranged from 2b–3b. After analysis, we could construct a timeline of physiological degeneration from the age of 20 until 50. A shift in biochemical composition of the IVD and small structural changes can be detected in this timeline. The loss and breakdown of proteoglycan (PG) appears to have a primary role in this initial stadium of LIDD. After the age of 50, degeneration accelerates, resulting in more ultrastructural changes of the IVD as well as loss of height, volume, and disc convexity. No significant difference in degree of LIDD was found between men and women. Finally, the lower lumbar levels L4/5 and L5/S1 had a significantly higher degree of LIDD than upper lumbar levels (L1/2, L2/3, and L3/4). *Conclusions*: This systematic literature review conceptualizes the theory of physiological ageing in the lumbar IVD based on MRI. Initially, there is change in biochemical composition of the IVD, which eventually results in ultrastructural changes. Future research should aim to validate this theory, preferably in a prospective cohort study.

## 1. Introduction

The vertebral column has an essential and central role in stability and daily activity. It is the central weight-bearing structure of the skeleton. The most important functions of the vertebral column are protection of the spinal cord and spinal nerves, support to the trunk, and providing flexibility to the body [1]. Its dysfunction can be extremely debilitating, for example in lower back pain (LBP). The intervertebral disc (IVD) allows movement between vertebral bodies and thus aids in the mobility of the spine. Some studies suggest that pain, specifically in the lower back, correlates to degeneration of the IVD [2,3,4]. The IVD therefore plays an essential role in the functioning of the spine.

The IVD has three main components: the nucleus pulposus (NP), annulus fibrosus (AF), and two hyaline cartilage endplates (CEP) [5,6]. The NP consists primarily of type II collagen and proteoglycan [6,7]. The negative charge of proteoglycan in the NP binds water. The NP consists of 66–86% water [5], which leads to swelling and subsequent pressure in the NP that is opposed by tensile forces in the AF [8,9]. This provides the NP with osmotic properties to resist compressive loads.

The AF consists of lamellae, which are mostly comprised of collagen type I but also contain proteoglycan, glycoproteins, and elastic fibres. [5,6,10]. The NP distributes applied loads to the more rigid AF, which is the main load-bearing structure of the IVD [6,10].

The CEP is a thin layer of hyaline cartilage between vertebral bone and soft disc tissue, which absorbs mechanical pressure and prevents the NP from bulging into vertebral bone. Furthermore, it allows nutrients to diffuse from the adjacent vertebral bone into the disc [11,12]. This is of importance since the intervertebral disc is largely avascular. Nutrients reach the NP and AF by diffusion through the extracellular matrix (ECM) [5,13]. The avascular nature and low amount of ECM producing connective tissue cells, such as chondrocyte-like cells, result in a low regenerative ability of the NP [5,14].

With aging, changes may occur in IVD that could be characterized as physiological. These changes have been identified in previous studies using histology. However, imaging, and particularly MRI, is also known to adequately distinguish between different structures of the IVD. With the use of various MRI modalities, it should be possible to determine these changes in the IVD not only in vitro but also in vivo [15,16]. If physiological aging of the IVD can be defined using MRI in healthy individuals, we could possibly identify abnormal (pathophysiological) aging of the IVD in symptomatic patients [17,18,19]. Identification of abnormal aging with MRI could aid in expanding our knowledge about diseases of the IVD and symptoms that have been associated with lumbar intervertebral disc degeneration (LIDD). Hopefully, this will result in a better understanding of patients’ symptoms and adequate treatment options.

This literature review therefore aims to establish an overview of the physiological age-related degenerative process of the lumbar IVD seen on MRI in an asymptomatic population.

## 2. Materials and Methods

### 2.1. Search Strategy and Study Selection

This systematic review was conducted using the Prisma guidelines [20]. This systematic review was not prospectively registered. The searches were constructed in collaboration with a medical information specialist (see acknowledgements). The searches were conducted in following databases: PubMed, Embase, Web of Science, and Cochrane. The main search terms were ‘(Inter)vertebral disc,’ ‘Magnetic resonance imaging’, and ‘(Inter)vertebral disc degeneration’. Detailed search strategies are available in Appendix A. The last search was run on 22 May 2023. After the search, duplicates were removed. Afterwards, studies were screened on title and abstract (MV). In case of doubt, titles and abstracts were discussed with the co-assessor (IC). Finally, full texts were screened and reviewed to determine inclusion (MV). The following eligibility criteria were used: focus on LIDD, results based on MRI, and only healthy individuals. Studies with patients suffering from LBP, patients who had undergone spinal surgery, or studies that focussed on the cervical or thoracic spine were excluded. Cadaver experimental studies were included; animal studies were excluded. In case of doubt, full texts were screened and discussed with the co-assessor (IC). Discrepancies between authors on final inclusion were discussed until consensus was reached.

### 2.2. Risk of Bias and Quality Assessment

One author (MV) assessed the quality of the included studies. The scoring tool of the Oxford Centre for Evidence-based Medicine was applied to determine the level of evidence. Included studies were scored from level 1 (systematic review) to 5 (expert opinion) based on study design [21,22]. The Cochrane ROBINS-I tool was used to determine the level of bias for seven domains in non-randomized controlled trials [23]. The studies were scored at low, moderate, serious, or critical risk of bias. When information to score a domain was not available, it was scored as ‘no information.’ The final score given to an article was equal to the worst given score in any of the seven domains.

### 2.3. Data Collection

The data were collected by one author (MV) using a data collection sheet. The following data were considered and collected: title, country, year of publication, study design, prospective/retrospective, correlation with age (including statistical methodology), ROBINS-I risk of bias, quality assessment, total study population, gender, age, levels of lumbar spine included, cadaver study (yes/no), main results, MRI-modality, magnetic field strength, and MRI-sequence. After data collection, all collected data were checked by the co-author (IC).

Interpretation of conflicting findings among the included studies were always discussed between authors (MV/IC). Greater weight was given to studies with larger sample sizes, those using validated MRI techniques, and those with a lower risk of bias (ROBINS-I, Appendix A).

## 3. Results

### 3.1. Study Selection

Results of the study selection process are illustrated in Figure 1. Searches in four databases identified 4756 eligible studies. After removal of duplicates, 2754 studies were left for screening of title and abstract. In total, 169 studies were selected for full text screening. Of 169 studies, 20 could not be retrieved. The remaining 149 articles were analysed, leading to the exclusion of 111 articles; 50 articles did not report data from healthy individuals and LBP patients separately, 45 articles had a focus other than LIDD, 7 articles were written in Chinese, 6 articles did not mention age, 2 articles had a focus on the thoracic spine, and 1 article was an animal study. This resulted in 38 studies to be included in the review.

### 3.2. Study Characteristics

Characteristics of the included studies are listed in Appendix A. Most studies were non-consecutive (n = 36), and two were exploratory cohort studies. Twenty-two studies were prospective and the remaining sixteen were retrospective. The publication years ranged from 2004 to 2022, and most studies were performed in the USA. All studies reported data based on different MRI techniques used to examine varying degrees of LIDD and IVD characteristics. Ten studies were cadaver-based (in vitro), one included both cadavers and living patients, and the remaining studies involved patients in vivo.

### 3.3. Risk of Bias and Quality Assessment

Seventeen studies had a low risk of bias Appendix A. There were 14 studies with a moderate risk of bias. In most cases, this was a result of possible bias in the selection process or bias induced by confounding, as these studies did not present clear selection criteria. Finally, there were seven studies with a serious risk of bias. In all cases this was a result of the possible risk of confounding due to a lack of patient characteristics reported in the studies. Most of the studies were level 3b evidence (n = 36), except for two studies with level 2b evidence [9,24].

### 3.4. Study Results

All results are shown in the Appendix A. In total, 1983 healthy individuals and 110 cadavers were included. Age ranged from 11 to 91 years. The smallest study had a population of four individuals and the largest included 627. Magnetic field strength expressed in Tesla (T) ranged from 1.0 T to 9.4 T, with the majority of the studies using 1.5 T or 3.0 T. Twenty-nine studied levels L1/L2 to L5/S1, three studied T12/L1 to L5/S1 [25,26,27], one studied T10/T11 to L5/S1 [28], one studied L2/L3 to L5/S1 [29], one studied L4/L5 to L5/S1 [30], one only studied L2/L3 [31], one only studied L4/L5 [32], and one did not report what part of the lumbar spine was studied [33].

Eleven different MRI-modalities were used to look at a variety of non-invasive biomarkers. Most studies used T1ρ mapping, T2 (relaxation) mapping, or diffusion weighted imaging (DWI) MRI sequences. Other MRI sequences used include 23 Na-Imaging, diffusion tensor imaging, magnetization transfer, Ultrashort time echo (UTE), Proton density-weighted spin echo (SE), Gadolinium enhanced T1 relaxation times, stimulated echo (STE), 3D UTE-Adiab-T1ρ, glycosaminoglycan chemical exchange saturation transfer (gagCEST), and water saturation shift referencing (WASSR).

Except for ten studies, all studies compared their results to the degenerative grade based on the Pfirrmann classification system using T2-weighted scans. Of the other ten, nine studies did not use a reference standard [9,24,31,34,35,36,37,38,39] and one study used the Thompson grading system [40].

Out of the 38 included studies, 21 studies looked directly at changes seen in biochemical composition, disc shape, location of degeneration, and/or the effect of gender on degeneration with advancing age. Other studies provided information about this for a certain age group. These results are listed in the following paragraphs.

Based on these results, this systematic literature review compiled literature on the physiological ageing process of the lumbar IVD that can be seen on MRI. The data were used to conceptualize the physiological ageing process of the lumbar IVD in a healthy population. A schematic representation of this timeline is presented in Figure 2 and will be elaborated on in the discussion.

### 3.5. Effect of Age on Degeneration

Out of the 21 studies that investigated the relation between age and LIDD seen on different MRI-modalities, 20 studies found a correlation between increased LIDD with older age in a linear or non-linear pattern. One study found no positive correlation between age and LIDD [25]. Twenty studies analysed the correlation between age and LIDD using linear regression, multiple regression, Pearson correlation coefficient, or the Spearman’s rank correlation coefficient [9,25,27,30,31,33,35,36,37,38,39,40,41,42,43,44,45,46,47,48]. These results are shown in Table 1.

Eight studies looked at the effect of age after subdivision into age groups [9,28,30,36,37,40,47,48]. These results are presented in Table 2. The different MRI modalities used by the included studies can also be found in Table 1 and Table 2.

### 3.6. Effect of Gender on LIDD

Twelve studies looked at the effect of gender on the degree of LIDD. Ten studies reported no significant difference between male and female gender [27,28,29,35,37,38,39,44,46,48], one study reported that males were more at risk [25], and one study reported that females were more at risk for LIDD [49].

### 3.7. Effect of Spinal Level on Degeneration

Eleven studies reported significantly increased LIDD at the lower (more caudal) lumbar discs levels (L4/L5–L5/S1) in comparison to the higher lumbar levels or in a linear correlation from L1/L2 up to L5/S1 [27,29,34,35,36,37,44,47,48,49,50].

## 4. Discussion

Ageing of the lumbar IVD occurs in the entire population. However, the process of the physiological ageing as seen on MRI has not been previously described. The literature compiled by this review will be discussed, proposing a theory of physiological ageing of the lumbar IVD. The characteristics of physiological ageing will be addressed per age group.

### 4.1. Zero to Twenty Years

No real conclusion about the biochemical content or structure of the disc can be made for age 0–20. Only one study included a small paediatric population and concluded that there was no difference in the apparent diffusion coefficient (ADC) in age zero to 20. Changes in ADC reflect changes in disc integrity, making it likely that disc integrity remains similar during this age [40]. Future research should try to include patients from an early age onwards.

### 4.2. Twenty to Fifty Years

Two studies suggest that at the age of 20 there is high water and proteoglycan matrix at the centre of the NP, and more collagen and less water towards the periphery [29,34]. With the progression of age from 20 onwards, changes in biochemical composition of the IVD start taking place. Six of the included studies studied non-invasive biomarkers sensitive to the proteoglycan matrix and suggested the breakdown and loss of proteoglycan from the NP [36,37,41,42,47,48]. Proteoglycan matrix consists, for a large part, of glycosaminoglycans. A reduction of biomarkers sensitive to glycosaminoglycan content (T1-rho mapping) in the NP was reported by two studies, indicating that there is a loss of proteoglycan [32,46].

The negative charge of proteoglycan attracts water through osmosis into the NP and causes swelling responsible for absorbing shocks and dividing pressure within the IVD [5,8,9]. Therefore, the loss of proteoglycan and glycosaminoglycans also induces the loss of water from the NP and progressive dehydration of the IVD from this moment onwards, as there are fewer binding locations for water. Five of the included studies reported progressive loss of water content from approximately 20 years of age onwards [9,39,41,44,51]. This leads to the decline of load-bearing properties of the disc, resulting in further damage of the proteoglycan matrix of the NP, as well as the collagen matrix [9,48]. This is supported by three of the included studies, suggesting that as proteoglycan matrix and glycosaminoglycan content of the NP changes, increasing signs of degeneration, such as small annular tears and loss of signal intensity in the NP, become visible on T2-weighted MRIs. With the progression of age, the frequency of these visible ultrastructural changes increases [32,35,43].

Due to the loss of water, the consistency of the IVD shifts from a more gelatinous nature to a more fibrous consistency with increasing collagen density. This results in a change in the ratio between collagen and water. This ratio shift was seen with T2 relaxation mapping in two studies [36,51]. Less water means that there is an increased collagen density in the IVD. This reduces the diffusivity in the IVD and may contribute to LIDD [31,51]. It is important to note that, during the whole degenerative process, the total collagen content of the IVD does not change; only the density changes due to the loss of water [36,40,51].

### 4.3. Fifty Years

From the age of 50, significant changes in MRI characteristics are seen that are suggestive of an increase in degeneration [31,36,47]. The first possible explanation for this is that early LIDD, characterized by a reduction of proteoglycan matrix and water in the NP before the age of 50, may reach a tipping point after the age of 50 [47]. Second, according to Wang et al. (2017), another explanation for the acceleration could be a loss of the regenerative capacity of the IVD. Measurements by Antoniou et al. (2004) also suggest that regenerative attempts during the degenerative process in the NP are made, suggesting that the IVD does have regenerative capacity [40,47]. Finally, the result of degeneration is the increased frequency of ultrastructural changes of the IVD [35,38,52]. Total proteoglycan and water content also keeps decreasing with time at an older age [33,50,53]. Furthermore, as a result of progression of LIDD with increasing age, the disc height and volume decrease slowly, as well as a loss of disc convexity, making degeneration visible even on plain radiographs [27,28,38].

### 4.4. CEP

On MRI, the CEP has a convex shape towards the vertebrae [24]. The CEP also appears as a bilaminar structure with a thicker inner layer of uncalcified CEP (on the side of the IVD) and a thinner outer layer of calcified CEP (adjacent to the vertebral body) [54]. The average CEP thickness decreases with age [12,45]. Wang et al. (2021) found that CEP T2 relaxation times correlate with the grade of degeneration and that the CEP plays a role in the LIDD process. Their results suggested that lower water content of the IVD causes solutes to diffuse less freely through the CEP and that increased collagen density hinders solute uptake by reducing pore space in the CEP [30]. On the other hand, two studies suggest that in degenerated discs, there are microfractures in the CEP that allow neovascularization and increased diffusion. Although there is no consensus on the precise role of the CEP in degeneration, in both cases a change in disc homeostasis occurs [41,55].

### 4.5. Interindividual Variability

Three studies noted that there were large interindividual differences in proteoglycan matrix of the NP in the same age group. This variation is likely to be a result of differences in biochemical content of the IVD and is suggested to be indicative of early signs of LIDD in some of these individuals [26,34,49]. Furthermore, variability detected by MRI-sequences such as DWI indicates heterogeneity of biochemical composition and is able to detect subtle changes that indicate early LIDD [56]. This emphasizes the importance of standardizing MRI sequences and reference ranges specific to the population in future research.

### 4.6. Location of Degeneration

While studying the degenerative process on the lumbar spine, twelve of the included studies noted a difference in degree and pattern of degeneration per lumbar level. They either described increased degeneration at lower lumbar levels L4/5 and L5/S1 compared to the upper lumbar levels L1/2, L2/3, and L3/4, or they described increased degeneration per level from the highest level L1/2 to the lowest level L5/S1 [27,29,34,35,36,37,44,47,48,49,50]. Three of these noted signs of early LIDD in the lower levels L4/5 and L5/S1 [34,36,48]. One study looking at the CEP reported different diffusion patterns in the CEP at lower lumbar levels [57].

These variations are possibly caused by differences in biomechanical loading between the lower and upper levels of the lumber spine. Interestingly, a recent study by Jamaludin et al. (2023) reported increased degeneration in the more caudal levels of LBP patients compared to a healthy control group [58]. This suggests that spinal level is important in LIDD and therefore should be considered in the description of physiological degeneration of the lumbar IVD.

### 4.7. Pfirrmann Grading

The theory on physiological degeneration of the IVD presented by this systematic literature review suggests that there are changes in the biochemical composition of the IVD before structural changes appear. This means that the Pfirrmann scoring tool for disc degeneration, which is not based on hard cut-off values, but subjective changes seen on T2-weighted MRIs, is not sensitive to early signs of LIDD. This is also suggested by ten of the studies included in this literature review. This highlights the need to develop a grading tool sensitive to early degenerative changes within the IVD, preferably an objective one [9,33,34,47,48,49,50,55,59,60].

### 4.8. Study Heterogeneity

The heterogeneity among studies results in considerable variation in MRI modalities, field strengths, and sequences. In addition to this, included studies have small sample sizes and are heterogenous in design. These factors collectively limit cross-study comparison and therefore sensitivity and subgroup analysis. On the other hand, the heterogeneity of this review allows for increased generalizability. The variation in MRI modalities and sequences highlights the need for standardized scanning protocols. T1ρ mapping, T2 (relaxation) mapping, or diffusion weighted imaging (DWI) are the most used and promising MRI sequences. Protocols combining these modalities with standardized acquisition parameters, field strengths, and outcome definitions are still needed. Future research should aim to validate combined modalities as this will improve reproducibility and allow comparison between studies.

### 4.9. Limitations

First, all the included studies have small sample sizes. Therefore, outcomes reported might not always be representative for the investigated group. This may be attributed to interindividual variation and possible confounding factors (e.g., BMI, geographical population, working class). Due to these small sample sizes, sufficient stratification and ability to perform subgroup analysis for potential confounding factors was lacking. Future research should take subgroup analysis into account when determining the appropriate sample size.

Second, this study is based on the assumption that ‘healthy’ subjects, without complaints of LBP, for example, portray physiological LIDD on MRI. Besides that, not all included studies clearly defined healthy subjects. However, this assumption had to be made as this is currently the best available option to determine physiological degeneration. A potential heterogeneity is therefore inherently introduced to the results and conclusion, underlining the need for clear definitions and inclusion criteria of healthy subjects in future research.

Part of the cadaver studies in this systematic review reported that they included healthy study subjects. However, we cannot completely rule out the presence of pathological degeneration in the cadaver studies. The main objective was to create an overview of all currently available literature, including cadaver studies. Since we do not know if the IVD changes can be reliably determined post-mortem on MRI scans, future research should investigate this.

### 4.10. Future Perspectives

To our knowledge, there is no research compiling data to determine the physiological ageing process of the IVD based on novel MRI-sequences rather than conventional T1 and T2 imaging. A longitudinal prospective cohort study, following healthy patients from a young age onwards, could help us to fill these knowledge gaps. Another option would be to perform a cross-sectional cohort study including individuals from all ages. Identification of the physiological aging process of the IVD using MRI is a prerequisite to identify pathophysiological degeneration patterns. This would be most important in diseases in which the pathophysiology is not yet completely understood, for example chronic low back pain.

## 5. Conclusions

This systematic literature review compiled literature to conceptualize a theory on the physiological ageing of the lumbar IVD seen on MRI. From the age of twenty until the age of fifty there is a suggested shift in biochemical composition of the IVD and small structural changes start to show. The loss and breakdown of proteoglycan appears to play a vital role in this initial stadium of LIDD. After the age of 50, changes in biochemical composition accelerate, resulting in ultrastructural changes of the IVD. This results in anatomical changes of the IVD such as loss of height, volume, and disc convexity. Future research should aim to look at physiological ageing in a study population from an early age onwards in larger prospective cohorts.

## Figures and Tables

**Figure 1 medicina-61-01430-f001:**
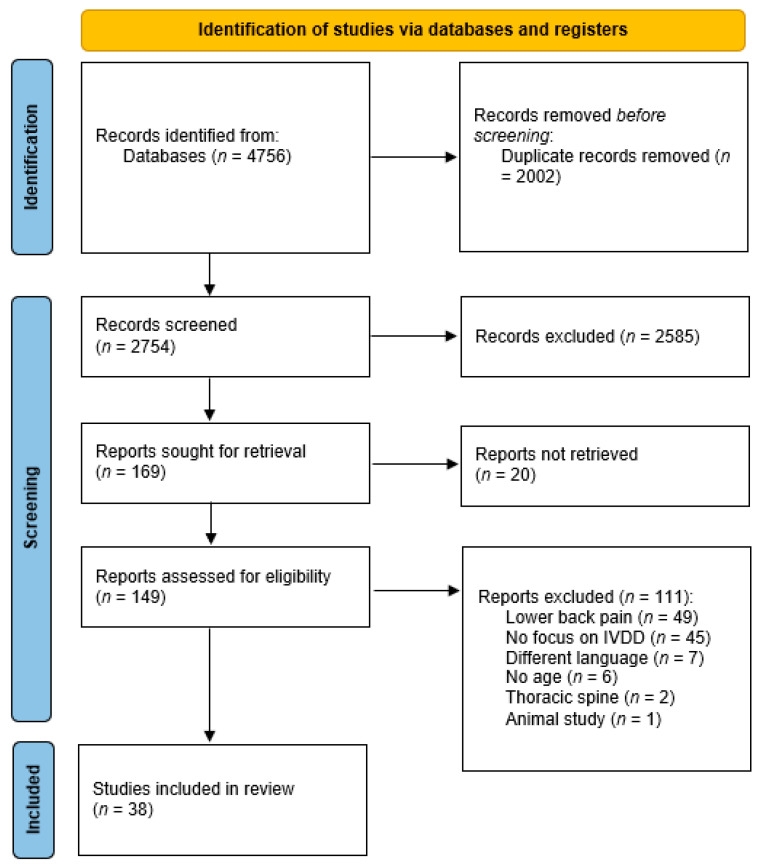
Prisma flow diagram of study selection.

**Figure 2 medicina-61-01430-f002:**
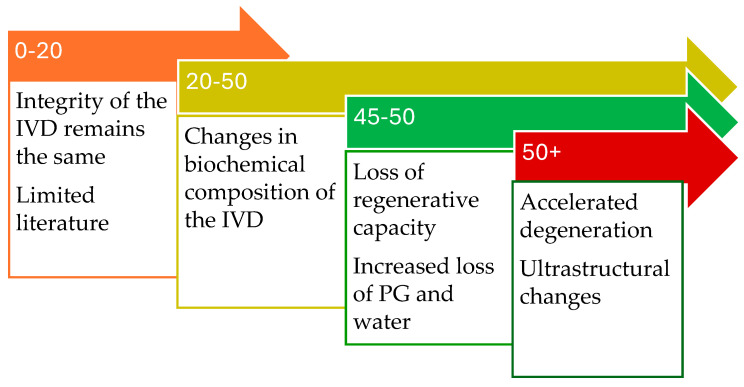
Summary of the suggested physiological ageing process of the lumbar IVD. (IVD = Intervertebral disc; PG = Proteoglycan).

**Table 1 medicina-61-01430-t001:** Results of studies analysing LVDD without subgroup division by age.

Author (Year)	Age	Results	Statistical Methodology
**Alkalay et al. (2018)** [31]	37–81 years	Independent of region, age was negatively associated with mean apparent diffusion coefficient (ADC) (r = −0.58, *p* < 0.001).	Linear correlation coefficient
**Antoniou et al. (2004)** [40]	11–77 years (mean 48)	The ADC in all directions decreased with age in the NP (ADCx: *p* < 0.0005; ADCy: *p* < 0.0001; ADCz: *p* < 0.002; ADCmean: *p* < 0.0001). The ADC in all directions decreased with age in the AF, except for ADCx (ADCy: *p* < 0.0001; ADCz: *p* < 0.002; ADCmean: *p* < 0.03).	Linear correlation coefficient
**DeLucca et al. (2016)** [45]	42–75 years (mean 60.9 ± 10.2)	The CEP anterior/posterior thickness and CEP average thickness decreased with age (r = −0.40, *p* < 0.01; r = −0.44, *p* < 0.01).	Multiple regression
**Filippi et al. (2013)** [48]	21–60 years	Statistically significant moderate negative correlation between average T1ρ values and age (r = 0.686, *p* < 0.01)	Spearman
**Fyllos et al. (2018)** [27]	18–54 years (mean 33.08)	Age was a significant coefficient for LVDD (*p* < 0.001).	Multiple regression
**Gübitz et al. (2018)** [37]	20–80 years	The effect of age on T1ρ per level was moderate to strong (L1/2 = −0.650; L2/3 = −0.698; L3/4 = −0.786; L4/5 = −0.770; L5/S1 = −0.589).	Pearson
**Haneder et al. (2014)** [25]	21–60 years (mean 29.2 ± 8.5)	Age had no or only a weak correlation to 23 Nanorm for all anatomic levels L1/2-L5/S1 (0.007 < R2 < 0.202).	Pearson
**Johannessen et al. (2006)** [33]	15–81 years (mean 51.6)	Strong correlation between T1ρ and age (r = −0.76, *p* < 0.01).	Linear correlation coefficient
**Matsumoto et al. (2013)** [35]	Mean age 48.0 ± 13.4	A decrease in disc signal intensity was significantly associated with an increase in age (odds ratio (OR) = 4.2; 95% confidence interval (CI) 1.2–14.9; *p* = 0.024).	McNemar’s test
**Menezes-Reis et al. (2016)** [39]	20–40 years (mean 27.1 ± 4.8)	Negative correlation between age and T2 relaxation time (r = −0.30, *p* < 0.0001)	Spearman
**Nguyen et al. (2008)** [41]	15–79 years (mean 51.8)	Negative correlation between T1ρ relaxation time and age (r = −0.84, *p* < 0.05).	Pearson
**Niu et al. (2011)** [46]	21–73 years (mean 40)	T2 exhibited a more significant inverse correlation with age than ADC (r = −0.77, *p* < 0.01; r = −0.37, *p* < 0.01).	Pearson
**Pfirrmann et al. (2006)** [38]	20–78 years	Age had a significant effect on disc height and disc volume (both *p* < 0.01).	Multilevel regression analysis
**Schleich et al. (2016)** [43]	21–49 years (mean 31 ± 8)	Significant correlation between age and morphological Pfirrmann classification, and age and CTF classification (r = 0.3175, *p* < 0.0001; r = 0.2476, *p* < 0.0001).	Pearson
**Vadapalli et al. (2019)** [9]	29–69 years	Strong correlation between fractional anisotropy (FA) (in the NP) and age (R2 = 0.6143).	Pearson
**Wang et al. (2017)** [47]	20–76 years (mean 34.2 ± 14.0)	T1ρ, T2 and ADC values decrease with the increase of age (r = −0.349; r = −0.594; r = −0.387; all *p* < 0.01).	Spearman
**Wang et al. (2021)** [30]	25–73 years (mean 36.9 ± 10.9)	Age is inversely associated with both mean T1ρ values in the NP and mean T2* values in the central CEP (r = −0.72, *p* < 0.001; r = −0.45, *p* = 0.032).	Pearson
**Wei et al. (2022)** [42]	25–71 years (mean 43 ± 16)	T1ρ values of the outer anterior AF and outer posterior AF showed positive correlations with age (r = 0.52; r = 0.71). The T1ρ value of the NP showed an inverse correlation with age (r = −0.76).	Spearman
**Zhang et al. (2012)** [36]	25–67 years (mean 46.8 ± 16)	Age had a negative correlation with mean diffusivity (MD) and moderately positive correlation with FA (r = −0.72, *p* < 0.001; r = 0.45, *p* < 0.001).	Spearman
**Zhang et al. (2014)** [44]	20–59 years (mean 41 ± 12)	Age had an inverse correlation with ADC for all spinal levels (L1/2 r = −0.381; L2/3 r = −0.518; L3/4 r = −0.537; L4/5 r = −0.576; L5/S1 r = −0.604; all *p* < 0.001).	Spearman

(ADC = apparent Diffusion Coefficient; AF = annulus fibrosis; CEP = cartilage endplate; CTF = combined task force classification; FA = Fractional anisotropy; NP = nucleus pulposis; T2* = T2 relaxation times).

**Table 2 medicina-61-01430-t002:** Results of studies analysing LVDD using subgroup division by age.

Author (Year)	Age	Results
**Antoniou et al. (2004)** [40]	11–77 years (mean 48) (subgroup analysis: 0–20; 21–40; 41–60; 61–80)	The ADC showed a significant decrease with older age when comparing different age groups in the NP and in the anterior AF, except no significant difference was found between age 0–20 and 21–40 as well as 41–60 and 61–80 (*p* < 0.002 0–20 compared with ages 41–60 and 61–80, *p* < 0.01 with ages 41–60 and 61–80).
**Filippi et al. (2013)** [48]	21–60 years (analysis per decade)	Statistically significant difference in the T1ρ values between all age groups (*p* < 0.01). No significant difference in T1p values between groups for 40–49 and 50–59 years.
**Gübitz et al. (2018)** [37]	20–80 years (subgroup analysis: A: 20–39; B: 40–59; C: 60–80)	Significant differences between groups A and C and groups B and C in T1p values (*p* = 0.0008; *p* = 0.0149).
**Machino et al. (2022)** [28]	Mean age 49.6 ± 16.5 (analysis per decade)	The intervertebral disk height at the anterior edge increased gradually with increasing age in both sexes. The intervertebral disk height at the center decreased gradually with increasing age in both sexes at the lower lumbar disks. The intervertebral disk height at the posterior edge did not remarkably change with age in either sex, except the height of the lower lumbar disks decreased.
**Vadapalli et al. (2019)** [9]	29–69 years (subgroup analysis: A = <30; B = 30–50; C = >50)	Subgroup A had a higher T2 and FA AF/NP ratio than subgroup B. However, the T2 of the AF was not significantly different. Subgroups A and B had a higher T2 and AF/NP ratio than subgroup C, but lower NP and AF values.
**Wang et al. (2021)** [30]	25–73 years (mean 36.9 ± 10.9) (subgroup analysis: <50; 51–60; >60)	NP T1ρ values were significantly correlated with CEP T2* values (r = 0.71, *p* = 0.047) in the youngest age group.
**Wang et al. (2017)** [47]	20–76 years (mean 34.2 ± 14.0) (analysis per decade)	T1ρ values remained relatively stable across the age range of 20–45 years and declined after the age of 45 years. T2 values were slowly decreased over the ages of 20–45 years, slightly increased over the age range of 45–50 years. Both values significantly declined after the age of 50 years (*p* < 0.01). The ADC values slowly decreased over the age range of 45-50 years. However, the ADC values fell rapidly after the age of 50 years (*p* < 0.05; *p* < 0.01).
**Zhang et al. (2012)** [44]	25–67 years (mean 46.8 ± 16) (subgroup analysis: 25–48; 53–67)	The FA shows a significant mean increase for the elderly group compared to the young adult group. The MD demonstrated significant mean decrease for the elderly group compared to the young adult group (both *p* < 0.001). Mean FA increased at a greater rate after age 48 years MD decreased at a greater rate after the age of 48.

(ADC = apparent diffusion coefficient; AF = annulus fibrosis; CEP = cartilage endplate; FA = functional anisotropy; MD = mean diffusivity; NP = nucleus pulposus; T2* = T2 relaxation times).

## Data Availability

See Appendix A.

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
