# Peer review of "Physiological Ageing of the Lumbar Intervertebral Disc Based on Magnetic Resonance Imaging, a Systematic Literature Review"

_medicina, 2025, doi:10.3390/medicina61081430_

Round 1
Reviewer 1 Report
Comments and Suggestions for Authors
The topic is highly relevant to both the clinical and research communities. Understanding the physiological ageing of the lumbar intervertebral disc (IVD) through MRI plays a critical role in early diagnosis, treatment planning, and differentiating between normal ageing and pathological degeneration. The paper provides a valuable consolidation of existing knowledge in this area.While the paper summarizes key imaging features associated with disc ageing (e.g., disc height reduction, signal intensity changes, Pfirrmann grading), a deeper comparative analysis across studies—such as age-specific trends, correlation with biomechanical changes, or contrast with degenerative disc disease—would enrich the discussion.
Author Response
Comments 1: [The topic is highly relevant to both the clinical and research communities. Understanding the physiological ageing of the lumbar intervertebral disc (IVD) through MRI plays a critical role in early diagnosis, treatment planning, and differentiating between normal ageing and pathological degeneration. The paper provides a valuable consolidation of existing knowledge in this area. While the paper summarizes key imaging features associated with disc ageing (e.g., disc height reduction, signal intensity changes, Pfirrmann grading), a deeper comparative analysis across studies—such as age-specific trends, correlation with biomechanical changes, or contrast with degenerative disc disease—would enrich the discussion.]
Response 1: [We agree that a deeper comparative analysis across studies would enrich the discussion. As to age specific trends we have explained what we could in the current discussion based on the included studies, a more in-depth comparative analysis of smaller age ranges was not possible due to small sample sizes and heterogeneity of the included studies. The latter has also been addressed in an extra paragraph in the discussion. Furthermore, we could add the effect of disc degeneration on biomechanical changes or contrast the physiological degeneration with pathological. However, we feel that this would dilute the review and distract the reader from our main study aim, namely identifying the physiological degeneration of the IVD. Nonetheless we agree that in future research this should be addressed as already mentioned in our discussion.]
Reviewer 2 Report
Comments and Suggestions for Authors
Dear Author,
Manuscript sound good and still more interesting if comments rectified from author end. My suggestions to clarify are -
- clarify the statement "low amount of ECM producing cells" with the cells involved
- are there any IVD differences in different geographical population, BMI composition, and pattern of working population
- "Variability in MRI has also been associated with difference in disc degeneration by Beattie et al. (2008)". Please elaborate the variability associated
- request to elaborate the proportion of degeneration at each vertebral level & which lumbar vertebra showed early degeneration and which is least susceptible. Also explain the anatomical basis for such physiological changes

Author Response
Comment 1: [clarify the statement "low amount of ECM producing cells" with the cells involved]
Response 1: [We acknowledge that this sentence lacked clarity. Accordingly, we have revised it in the introduction and trust that it now communicates our intended meaning more effectively.]
Comment 2: [are there any IVD differences in different geographical population, BMI composition, and pattern of working population]
Response 2: [This is a valid comment, however due to the small sample size of available studies we could not perform a subgroup analysis, as it would have been of limited value. Therefore, we have added it to the limitations section of the discussion.]
Comment 3: ["Variability in MRI has also been associated with difference in disc degeneration by Beattie et al. (2008)". Please elaborate the variability associated]
Response 3: [This sentence does indeed lack context; we have elaborated onto this in the discussion. We hope that the provided context sufficiently explains the associated variability.]
Adjusted text 3: [“Furthermore, variability detected by MRI-sequences such as DWI, indicates heterogeneity of biochemical composition and is able to detect subtle changes that indicate early LIDD. This emphasizes the importance of standardizing MRI sequences and reference ranges specific to the population in future research.”]
Comment 4: [request to elaborate the proportion of degeneration at each vertebral level & which lumbar vertebra showed early degeneration and which is least susceptible. Also explain the anatomical basis for such physiological changes]
Response 4: [The levels which show signs of early degeneration have also been added under subheading ‘location of degeneration’ in the discussion. Unfortunately, the proportion of degeneration for every level is not available; Due to the small sample sizes of the included studies this level of subgroup analysis could not be performed. Furthermore, we have considered adding biomechanical explanations to the physiological changes but feel that it would dilute the review and distract the reader from our main study aim.]
Reviewer 3 Report
Comments and Suggestions for Authors
Thank you for submitting your papers to the Medicina.
Here are my propsed recommendations for improvement of manuscript.
1) The authors should expand their discussion of study heterogeneity and its impact on conclusions. If possible, I recommend to authors doing sensitivity analyses or subgroup analysis by MRI technique.
2) Discuss how varying definitions of healty subjects acorss studies may affect conclusions.
3) Dicuss more specific recommendations for standardized MRI protocols for future studies.
4) Clarify methodology for handling conflicting results between studies
5) Would you Improve presentation of Tables 1 and 2 for better readability?
Comments on the Quality of English Language
acceptable
Author Response
Comment 1: [The authors should expand their discussion of study heterogeneity and its impact on conclusions. If possible, I recommend to authors doing sensitivity analyses or subgroup analysis by MRI technique.]
Response 1: [Based on your comment we decide to revise this topic and added a paragraph in the discussion. Here, the effect of study heterogeneity, its impact on our conclusions, and recommendations for future research are discussed. See subheading ‘Study heterogeneity’ in the discussion for the full paragraph.]
Comment 2: [Discuss how varying definitions of healthy subjects across studies may affect conclusions.]
Response 2: [This is a very valid point, we have therefore decided to revise this in the limitations section of the discussion. We have elaborated on how varying definitions of healthy subjects across studies may affect our conclusion. Please have a look at the limitations section of the discussion for the full text.]
Comment 3: [Discuss more specific recommendations for standardized MRI protocols for future studies.]
Response 3: [See subheading ‘Study heterogeneity’. Here we explain the difficulties and need for standardized MRI protocols.]
Comment 4: [Clarify methodology for handling conflicting results between studies]
Response 4: [See methodology, subheading data collection. Here we inserted a paragraph clarifying how conflicting data was handled.]
Comment 5: [Would you Improve presentation of Tables 1 and 2 for better readability?]
Response 5: [The data in Tables 1 and 2 have been presented in this format as it was deemed most appropriate. Based on your suggestion, it is unclear what specific improvements the reviewer would recommend. We welcome any detailed suggestions for enhancement and will consider potential modifications accordingly.]